# Characterization of *mcr-1*-Harboring Plasmids from Pan Drug-Resistant *Escherichia coli* Strains Isolated from Retail Raw Chicken in South Korea

**DOI:** 10.3390/microorganisms7090344

**Published:** 2019-09-12

**Authors:** Jinshil Kim, Bo Kyoung Hwang, HyeLim Choi, Yang Wang, Sang Ho Choi, Sangryeol Ryu, Byeonghwa Jeon

**Affiliations:** 1Department of Food and Animal Biotechnology, Research Institute for Agriculture and Life Sciences, Center for Food and Bioconvergence, Seoul National University, Seoul 08826, Korea; jinsilk1130@naver.com; 2Department of Agricultural Biotechnology, Center for Food Safety and Toxicology, Seoul National University, Seoul 08826, Korea; rose0814p@naver.com (B.K.H.); helenchoi501@gmail.com (H.C.); choish@snu.ac.kr (S.H.C.); 3Food-borne Pathogen Omics Research Center (FORC), Seoul National University, Seoul 08826, Korea; 4Beijing Advanced Innovation Center for Food Nutrition and Human Health, College of Veterinary Medicine, China Agricultural University, Beijing 100083, China; wangyang@cau.edu.cn; 5Environmental Health Sciences, School of Public Health, University of Minnesota, Minneapolis, MN 55455, USA

**Keywords:** *Escherichia coli*, *mcr-1*, food, retail raw chicken

## Abstract

A number of studies from different countries have characterized *mcr-1*-harboring plasmids isolated from food; however, nothing has been reported about it in South Korea. In this study, we report the characterization of *mcr-1* plasmids from pan drug-resistant (PDR) *Escherichia coli* strains isolated from retail food in the country. Colistin-resistant *E. coli* strains were isolated from retail raw chicken, and PCR was carried out to detect the *mcr-1* gene. Whole genome sequencing of the *mcr-1*-positive strains was performed for further characterization. The results of whole genome sequencing revealed that all *mcr-1* plasmids belonged to the IncI2 type. In addition to the *mcr-1* plasmids, all of the isolates also carried additional plasmids possessing multiple antibiotic resistance genes, and the PDR was mediated by resistant plasmids except for fluoroquinolone resistance resulting from mutations in *gyrA* and *parC.* Interestingly, the *mcr-1* plasmids were transferred by conjugation to other pathogenic strains including enterohemorrhagic *E. coli* (EHEC), enterotoxigenic *E. coli* (ETEC), enteroaggregative *E. coli* (EAEC), *Salmonella*, and *Klebsiella* at the frequencies of 10^−3^−10^−6^, 10^−2^−10^−5^, 10^−4^−10^−5^, 10^−4^−10^−6^, and 10^−5^−10^−6^, respectively. The results showed that *mcr-1* plasmids can be easily transmitted to pathogenic bacteria by conjugation.

## 1. Introduction

The emergence of multidrug-resistant (MDR) pathogens, such as ESKAPE (*Enterococcus faecium, Staphylococcus aureus, Klebsiella pneumonia, Acinetobacter baumannii, Pseudomonas aeruginosa, Enterobacter spp.,* or recently *Enterobacteriaceae*) and the lack of effective antimicrobials are a serious issue in public health [1]. Colistin is one of the last-resort antibiotics to treat MDR Gram-negative bacteria, and its binding to the lipid A moiety of lipopolysaccharide (LPS) destabilizes the outer membrane and results in cell death [2]. Colistin resistance is mainly associated with the modification of LPS, such as phosphoethanolamine modification of lipid A [3,4,5], and causes serous clinical problems in the control of Gram-negative pathogens in ESKAPE [6,7,8].

Since the first discovery of the mobilized colistin resistance (*mcr*)*-1* gene on plasmids in *Escherichia coli* by Liu et al. in China, 2016 [4], *E. coli* strains harboring *mcr-1* have been reported in many countries throughout America, Asia, and Europe [4,9,10] and have been isolated from various sources, such as animals, humans, environmental samples, and food [10,11,12]. In South Korea, *mcr-1*-positive *E. coli* strains have been isolated in livestock and humans [13,14]. However, there have been no studies about *mcr-1-*positive *E. coli* from retail food in the country.

Several recent reports also suggested that the spread of *mcr-1* to multidrug-resistant bacteria can contribute to the development of the pan drug-resistant (PDR) phenotype, since *mcr-1* on plasmids can be easily disseminated [3,15]. The increasing number of PDR bacteria, including *E. coli*, is considered a threat to public health [16]. *E. coli* is a major cause of human diseases, such as urinary tract infections, sepsis, and pneumonia [17]. Therefore, the emergence of PDR *E. coli* isolates, particularly those harboring extended-spectrum β-lactamase (ESBL) and plasmid-mediated quinolone resistance (PMQR) genes, have aggravated the public health burdens of antibiotic resistance.

A number of studies have shown that retail chicken is a major reservoir of disseminating antibiotic-resistant *E. coli* to humans [9,18,19]. In this study, we aimed at isolating *mcr-1*-positive *E. coli* strains from retail chicken in South Korea, characterizing the DNA sequence of the *mcr-1* plasmids, and determining the frequencies of conjugational transfer of *mcr-1* plasmids to other Gram-negative pathogens.

## 2. Materials and Methods

### 2.1. Bacterial Strains and Culture Methods

Three *mcr-1*-positive *E. coli* strains (JSMCR1, FORC81 and FORC82) were isolated from retail raw chicken in South Korea in our previous study [20]. The *mcr-1*-positive *E. coli* strains, *E. coli* ATCC 43889 (Enterohemorrhagic *E. coli*; EHEC), *E. coli* NCCP 14039 (Enteroaggregative *E. coli*; EAEC), enterotoxigenic *E. coli* (ETEC, a laboratory collection), *Salmonella enetrica* serovar Typhimurium SL1344, and *K. pneumoniae* (a laboratory collection) were cultured on Luria–Bertani (LB) media at 37 °C. The pathogenic *E. coli* strains (EHEC, ETEC, EAEC), *Salmonella* Typhimurium, and *Klebsiella* were used as recipient strains in the conjugation assay.

### 2.2. Antimicrobial Susceptibility Testing

Antimicrobial susceptibility testing was performed with a broth dilution method as described previously [21,22] with ten antibiotics, including ampicillin, cephalothin, tetracycline, chloramphenicol, ciprofloxacin, kanamycin, gentamicin, streptomycin, polymyxin B, and colistin. *E. coli* ATCC 25922 was used as the quality control strain.

### 2.3. Conjugation Assay

For the selection of transconjugants, we first obtained spontaneous mutants of streptomycin-resistant recipient strains by culturing them on LB media supplemented with streptomycin. Donor and recipient cells were prepared by transferring 1% inoculum from overnight cultures into fresh LB broth, followed by incubation at 37 °C for 4 h with constant shaking. *E. coli* was conjugated with recipient cells at a ratio of 1:1. Cells were pelleted by centrifugation, washed twice with 10 mM MgSO_4_, and resuspended in 50 µL of MgSO_4_. The mixture of donor and recipient cells were spread on LB agars supplemented with streptomycin (2 µg/mL) and colistin (4 µg/mL). Transconjugants were confirmed with PCR using *mcr-1*-specific primers [4] and the recipient strains. Conjugation frequencies were calculated as the number of transconjugants per recipient cell.

### 2.4. Whole-Genome Sequencing

Whole-genome sequencing and assembly were performed commercially at ChunLab Inc. (Seoul, South Korea). The whole genome of *E. coli* JSMCR1, FORC81 and FORC82 was sequenced using PacBio RS II (Pacific Biosciences, Menlo Park, CA, USA). The genome sequences were annotated using the online Rapid Annotation Subsequencing Technology (RAST) and CLC Main Workbench 3.6.1 (CLC bio, Aarhus, Denmark), and deposited in the GenBank database with accession numbers CP030152-CP030157 (JSMCR1), CP029057-CP029061 (FORC81), and CP026641-CP026644 (FORC82).

## 3. Results and Discussion

### 3.1. Whole-Genome Sequencing of mcr-1-Positive E. coli Strains

Three *mcr-1*-positive *E. coli* strains (JSMCR1, FORC81 and FORC82) were isolated from retail chicken in South Korea. The results of whole genome sequencing revealed that the three *mcr-1*-postive *E. coli* strains possessed multiple plasmids and some of the plasmids harbored a number of antibiotic resistance genes (Table 1 and Appendix A). The three *mcr-1*-harboring plasmids belonged to the IncI2 type and possessed the genetic elements for bacterial conjugation (Table 1). The three *mcr-1*-harboring plasmids were similar to pHNSHP45 (accession no. KP347127), the first *mcr-1*-harboring plasmid isolated in China; pJSMCR1_4 (96% query coverage, 100% max nucleotide identity), pFORC81_2 (89% query coverage, 99% max nucleotide identity), and pFORC82_3 (97% query coverage, 99% max nucleotide identity). Unlike pHNSHP45, however, insertion sequences were not found in the plasmids (Figure 1). The sequences of the *mcr-1*-harboring plasmids were similar to the IncI2-type *mcr-1*-harboring plasmids, which were isolated from livestock and humans in Korea [13,14]. This is also consistent with a previous extensive analysis revealing that IncI2 is predominant in *mcr-1*-harboring plasmids in Asia, whereas IncHI2 plasmids are predominant in Europe [23].

### 3.2. Antimicrobial Susceptibility Profiles and Other Resistance Genes

All of the *mcr-1*-positive *E. coli* strains were highly resistant to most of the tested antibiotics belonging to different classes (Table 2). In particular, *E. coli* JSMCR1 was resistant to all the antibiotics tested in this study. Based on the results of whole genome sequencing, all isolates carried a few plasmids with different replicon types and multiple other antibiotic resistance genes conferring resistance to several different antibiotic classes (Table 1 and Appendix A). Whole-genome sequencing discovered point mutations in *gyrA* and *parC*, which confer resistance to fluoroquinolones; however, other antibiotic resistance genes were not found in the chromosome of the three strains, suggesting that pan drug resistance is primarily mediated by resistance plasmids in the strains.

### 3.3. Conjugation Assay

The conjugation experiments were performed with pathogenic *E. coli* strains (EHEC, ETEC, EAEC), *Salmonella* Typhimurium, and *Klebsiella* as the recipient strains. The results of the antimicrobial susceptibility test showed that only the minimum inhibitory concentrations (MICs) of polymyxin B and colistin were increased in transconjugants from two-fold to 16-fold compared to their parental strains (Table 2). However, other plasmids harboring antibiotic resistance genes, which were simultaneously present in the *mcr-1*-postive strains, were not transferred to the recipient strains under the experimental settings based on PCR testing with primers specific to the plasmids. These results indicated that *mcr-1* plasmids are highly transmissible compared to other resistance plasmids; this presumably enables *mcr-1* plasmids to be disseminated from *E. coli* to Gram-negative bacteria [4,24].

To confirm this, we determined the conjugational frequency of *mcr-1* plasmids and found that the transmission rates of the *mcr-1* plasmids were 10^−3^–10^−6^ in EHEC, 10^−2^–10^−5^ in ETEC, 10^−4^–10^−5^ in EAEC, 10^−4^–10^−6^ in *Salmonella*, and 10^−5^–10^−6^ in *Klebsiella* (Figure 2). Conjugation frequencies varied depending on the recipient strain. For instance, the conjugation frequencies of pFORC81-2 were as high as 3 × 10^−3^ in ETEC and as low as 2 × 10^−6^ in EHEC (Figure 2). The frequencies of conjugational transfer of the *mcr-1* plasmids to *E. coli* strains were similar to previous studies [25]. Furthermore, in this study, we demonstrated that *mcr-1* plasmids were transmitted to other Gram-negative bacteria, such as *Salmonella* and *Klebsiella*, at the frequencies comparable to those observed in *E. coli* (Figure 2).

## 4. Conclusions

In this study, we isolated and characterized three *mcr-1* plasmids from PDR *E. coli* strains from retail raw chicken in South Korea. Although the number of isolated plasmids was small, the findings of this study are important because this is the first report about *mcr-1* plasmids originating from retail food in the country. The whole genome sequencing of the PDR *E. coli* strains showed that all the genetic determinants for antibiotic resistance were associated with plasmids, except for fluoroquinolone resistance caused by point mutations in *gyrA* and *parC*. The *mcr-1* plasmids were highly transferrable to pathogenic *E. coli* strains, *Salmonella*, and *Klebsiella*. This may allow colistin resistance to easily spread in the food supply system.

## Figures and Tables

**Figure 1 microorganisms-07-00344-f001:**
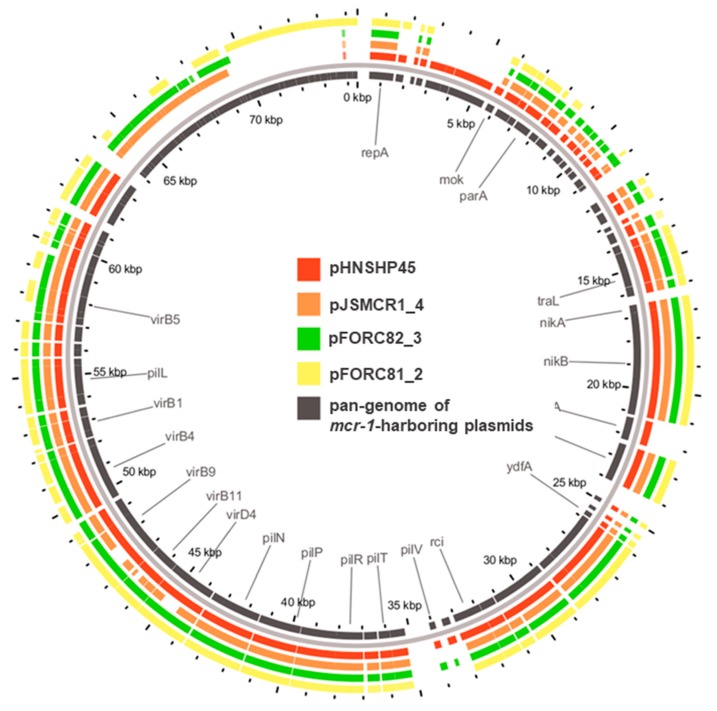
Sequence comparison of *mcr-1*-harboring plasmids. pHNSHP45 was used as a reference. Black inner ring indicated the pan-genome of the *mcr-1*-harboring plasmid.

**Figure 2 microorganisms-07-00344-f002:**
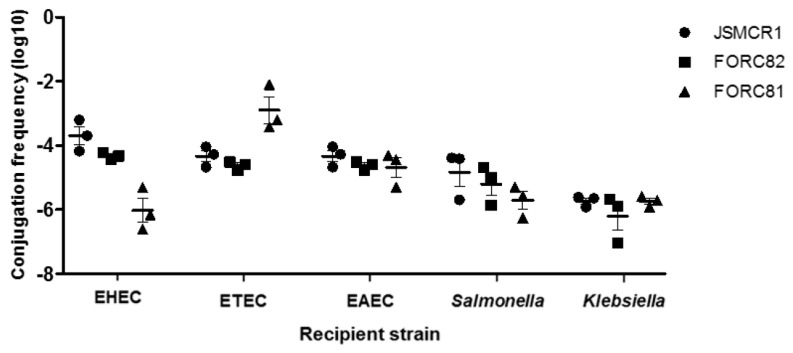
Conjugation frequencies of three *mcr-1* plasmids from *E. coli* isolates from retail chicken. The data represent the means and standard deviations of the results from three independent experiments.

**Table 1 microorganisms-07-00344-t001:** Plasmids present in *E. coli* isolates harboring *mcr-1.*

*E. coli* Strain	Plasmid	Size (bp)	GenBank Accession No.	Inc Group	Resistance Genes
JSMCR1	pJSMCR1_1	152,677	CP030153	IncFIB, IncFII	*aph(3′)-Ia, aac(3)-IId, bla_CTX-M-65_, fosA3*
	pJSMCR1_2	134,064	CP030154	p0111	*aadA1, blaOXA-10, qnrS1, floR, cmlA1, arr-2, tet(A), dfrA14*
	pJSMCR1_3	109,689	CP030155	IncI1	*-*
	pJSMCR1_4	61,828	CP030156	IncI2	*mcr-1*
	pJSMCR1_5	29,589	CP030157	IncX4	*-*
FORC81	pFORC81_1	253,947	CP029058	IncI1, IncFII	*aadA1, aadA2, aac(3)-IId, blaTEM-1B,* *qnrS1, floR, cmlA1, sul3, tet(A), dfrA12*
	pFORC81_2	61,917	CP029059	IncI2	*mcr-1*
	pFORC81_3	38,749	CP029060	IncX1	*-*
	pFORC81_4	32,945	CP029061	IncI1	*blaTEM-1B, floR*
FORC82	pFORC82_1	250,778	CP026642	IncHI2A, IncHI2, IncN	*aadA1, aph(3′’)-Ib, aph(6)-Id, bla_CTX-M-65_, bla_OXA-10_, qnrS1, mph(A), floR, cmlA1,* *arr-2, sul2, tet(M), tet(A), dfrA14*
	pFORC82_2	101,404	CP026643	IncFIC, IncFIB	*-*
	pFORC82_3	65,206	CP026644	IncI2	*mcr-1*

**Table 2 microorganisms-07-00344-t002:** Minimum inhibitory concentrations (MICs) of *mcr-1*-positive *E. coli* strains and their transconjugants.

Strain	Origin ^a^	*mcr-1* Gene ^b^	MIC ^c^ (μg/mL)									
			AMP	CEF	TET	CHL	CIP	KAN	GEN	STR	POL	COL
*E. coli* JSMCR1	WT	+	>64	>64	128	>64	>8	>32	>64	64	8	8
*E. coli* FORC81	WT	+	>64	64	>128	>64	>8	8	>64	8	8	8
*E. coli* FORC82	WT	+	>64	>64	128	64	2	4	2	2	8	8
EHEC (*E. coli* ATCC 43889)	WT	-	≤0.5	≤0.5	≤0.0628	≤0.5	≤0.0039	≤0.25	≤0.5	>128	≤0.25	≤0.25
pJSMCR1/EHEC	TC	+	≤0.5	≤0.5	≤0.0628	≤0.5	≤0.0039	≤0.25	≤0.5	>128	4	4
pFORC81/EHEC	TC	+	≤0.5	≤0.5	≤0.0628	≤0.5	≤0.0039	≤0.25	≤0.5	>128	4	4
pFORC82/EHEC	TC	+	≤0.5	≤0.5	≤0.0628	≤0.5	≤0.0039	≤0.25	≤0.5	>128	4	4
ETEC (isolate)	WT	-	>64	8	32	64	0.0312	>32	>64	>128	4	2
pJSMCR1/ETEC	TC	+	>64	8	32	64	0.0312	>32	>64	>128	8	8
pFORC81/ETEC	TC	+	>64	8	32	64	0.0312	>32	>64	>128	8	8
pFORC82/ETEC	TC	+	>64	8	32	64	0.0312	>32	>64	>128	8	8
EAEC (*E. coli* NCCP 14039)	WT	-	>64	16	>128	4	0.0312	8	2	>128	4	2
pJSMCR1/EAEC	TC	+	>64	16	>128	4	0.0312	8	2	>128	8	4
pFORC81/EAEC	TC	+	>64	16	>128	4	0.0312	8	2	>128	8	8
pFORC82/EAEC	TC	+	>64	16	>128	4	0.0312	8	2	>128	8	8
*S.* Typhimurium SL1344	WT	-	2	2	0.5	4	0.0156	4	2	>128	4	4
pJSMCR1/SL1344	TC	+	2	2	0.5	4	0.0156	4	2	>128	8	8
pFORC81/SL1344	TC	+	2	2	0.5	4	0.0156	4	2	>128	8	8
pFORC82/SL1344	TC	+	2	2	0.5	4	0.0156	4	2	>128	8	8
*Klebsiella* (isolate)	WT	-	>64	16	>128	>64	>16	>32	1	>128	4	4
pJSMCR1/*Klebsiella*	TC	+	>64	16	>128	>64	>16	>32	1	>128	8	16
pFORC81/*Klebsiella*	TC	+	>64	16	>128	>64	>16	>32	1	>128	8	16
pFORC82/*Klebsiella*	TC	+	>64	16	>128	>64	>16	>32	1	>128	32	16

^a^ WT: wild type, TC: Transconjugant. ^b^ Presence (+) or absence (-) of *mcr-1*, based on PCR and confirmed by sequencing. ^c^ AMP: ampicillin, CEF: cephalothin, TET: tetracycline, CHL: chloramphenicol, CIP: ciprofloxacin, KAN: Kanamycin; GEN: gentamicin; STR: streptomycin; POL: polymyxin B; COL: colistin.

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
