# Peer review of "Characterization of mcr-1-Harboring Plasmids from Pan Drug-Resistant Escherichia coli Strains Isolated from Retail Raw Chicken in South Korea"

_microorganisms, 2019, doi:10.3390/microorganisms7090344_

Round 1
Reviewer 1 Report
General comments
In their manuscript, the authors characterise three mcr-1-positive E. coli strains and the plasmids of these strains. The manuscript is short and informative:
Whole genome sequencing analyses revealed that pandrug resistance, with the exaction of fluoroquinolone resistance, was primarily mediated by resistance plasmids.
The conjugation experiments interestingly were performed with pathogenic strains and the mcr-1-harboring strains were shown to be more transmissible than other plasmids harbouring antibiotic resistant genes.
Conjugation frequencies for different pathogenic E. coli as well as Salmonelly and Klepsiella strains were determined.
The manuscript is worded in a brief and concise manner and therefore good to read. At some points, more detailed information would ease the understanding or is even necessary to ensure reproducibility.
Specific comments
Abstract
Please reword the abstract, particularly the results section (compare to the well written conclusion).
The introductory sentences are fine. In the second sentence (line 17 f) you anticipate the results. Line 22:
In my opinion it is not necessary to state the method the transconjugants were obtained with in the abstract à delete 'with a conjugation assay' (it is rather logical). Instead, it could be interesting to the reader to specify, that conjugation frequencies were determined. Line 25: Specify that 'pandrug resistance is primarily mediated by resistance plasmids' (copied from section 3.2) Line 26: Specify 'easily transferred to other pathogenic strains including …'. In my opinion, the last sentence should be removed or postponed. If you leave the second sentence of the abstract, you already have the information in there (don't mention it twice). If you remove the second sentence, please move this information at the beginning of the results part of the abstract because for me your other results are much more interesting.
Introduction
Line 33: Even though this might already have become common knowledge, please ad year and country to the Liu et al. citation.
Line 46 ff: The last paragraph of the introduction should be worded in such a way to state the aims of the study. Please reword.
Materials and Methods
Line 54 ff: Please indicate what these additional strains were used for (conjugation …).
Line 60 ff: Please state which substances have been tested. Possibly not all of them are listed in the results section. If so, please at least refer to the results section.
Results and Discussion
In this section, the discussion of possible limitations is missing. For example, the assembly of plasmids from whole-genome data can be complex. How do you ensure the accuracy of your results? Have you isolated and sequenced the plasmids themselves?
Please indicate limitations of the methods used and the conclusions drawn.
Line 82: The Results section starts somewhat sudden. Please ad an introductory sentence, e.g. explaining the different plasmids you have isolated and analysed.
Line 89: Change 'form' to 'from'
Figures and Tables
Figure 1: Revise the labelling (last sentence). 'The black inner ring indicates the pan-genome of mcr-1-harboring plasmids.'
Author Response
Thanks you for the constructive comments. Please read our responses below.
Abstract
Please reword the abstract, particularly the results section (compare to the well written conclusion).
The introductory sentences are fine. In the second sentence (line 17 f) you anticipate the results. Line 22: In my opinion it is not necessary to state the method the transconjugants were obtained with in the abstract à delete 'with a conjugation assay' (it is rather logical). Instead, it could be interesting to the reader to specify, that conjugation frequencies were determined.
Response: According to the comment, we deleted the method description in the abstract and added a statement about the results (Lines 28-30).
Line 25: Specify that 'pandrug resistance is primarily mediated by resistance plasmids' (copied from section 3.2)
Response: According to the reviewer’s comment, we included the sentence in the revised version (Lines 24, 25)
Line 26: Specify 'easily transferred to other pathogenic strains including …'.
Response: We rephrased the sentence by showing the conjugational frequencies in the revised version (Lines 25-30).
In my opinion, the last sentence should be removed or postponed. If you leave the second sentence of the abstract, you already have the information in there (don't mention it twice). If you remove the second sentence, please move this information at the beginning of the results part of the abstract because for me your other results are much more interesting.
Response: We deleted the last sentence in the abstract in the revised version.
Introduction
Line 33: Even though this might already have become common knowledge, please ad year and country to the Liu et al. citation.
Response: We mentioned it in the new version (Line 43)
Line 46 ff: The last paragraph of the introduction should be worded in such a way to state the aims of the study. Please reword.
Response: The last sentence was reworded (Lines 56-59).
Materials and Methods
Line 54 ff: Please indicate what these additional strains were used for (conjugation …).
Response: We indicated it in the revised version (Lines 125, 126).
Line 60 ff: Please state which substances have been tested. Possibly not all of them are listed in the results section. If so, please at least refer to the results section.
Response: We stated the substances in the revised version. (Lines 129-131)
Results and Discussion
In this section, the discussion of possible limitations is missing. For example, the assembly of plasmids from whole-genome data can be complex. How do you ensure the accuracy of your results? Have you isolated and sequenced the plasmids themselves? Please indicate limitations of the methods used and the conclusions drawn.
Profs. Ryu and Choi lead the 10K Foodborne Pathogen Genome Sequencing Project funded by the Korea Food and Drug Administration, and whole genome sequencing of a significant number (about 10,000) of pathogens is performed in the lab. Thus, we did not experience any particular difficulties in the study. Thank you for the comment.
Line 82: The Results section starts somewhat sudden. Please ad an introductory sentence, e.g. explaining the different plasmids you have isolated and analysed.
Response: Based on the comment, we revised the first part of the results (Lines 62-65).
Line 89: Change 'form' to 'from'
Response: We corrected it (Line 135)
Figures and Tables
Figure 1: Revise the labelling (last sentence). 'The black inner ring indicates the pan-genome of mcr-1-harboring plasmids.'
Response: We changed it in the revised version.
Reviewer 2 Report
The manuscript entitled “Characterization of mcr-1-harboring plasmids from 2 pandrug-resistant Escherichia coli strains isolated 3 from retail raw chicken in South Korea” by Kim et al. uses current methods for the characterization of colistin-resistant E. coli from retail foods in South Korea. The manuscript is very interesting, and it is adequately written, the grammar is also correct. The paper holds important information for an international readership, however, there are some points that need to be corrected, before the paper could be published. Please find my recommendations below:
In the first paragraph: Include a sentence that describes that the emergence of drug resistant pathogens and the lack of newly developed antimicrobial agents is a severe global public health issue.
Include the following publication as a reference:
https://www.mdpi.com/1420-3049/24/5/892
In the first paragraph, please briefly describe the mechanism of action for colistin, and the possible mechanisms of colistin-resistance. In addition, please describe the existence of other colistin-resistance plasmids.
Line 32: „multidrug-resistant (MDR) Gram-negative bacteria”.
Line 39-46: Describe the relevance of so-called “ESKAPE” bacteria, especially the Gram-negative constituents, where colistin-resistance is an important issue.
Include the following publications as a reference:
https://www.ncbi.nlm.nih.gov/pubmed/29310479
https://www.mdpi.com/2079-9721/7/2/41
https://www.ncbi.nlm.nih.gov/pubmed/19035777
Line 55: an unnecessary space after enteroaggregative
Results and discussion: there are three mcr-1 harboring strains tested in this study. Although this number is not that high, significant results were produced in the study. Please describe in the discussion that despite the small number of strains, why is this information novel and significant.
Author Response
Thank you for the constructive comments. Please read our responses below.
In the first paragraph: Include a sentence that describes that the emergence of drug resistant pathogens and the lack of newly developed antimicrobial agents is a severe global public health issue. Include the following publication as a reference: https://www.mdpi.com/1420-3049/24/5/892
Response: We included a new sentence in the revised version (Lines 34-37)
In the first paragraph, please briefly describe the mechanism of action for colistin, and the possible mechanisms of colistin-resistance. In addition, please describe the existence of other colistin-resistance plasmids.
Response: We described it in the revised version (Lines 37-41)
Line 32: „multidrug-resistant (MDR) Gram-negative bacteria”.
Response: We changed it (Line 34)
Line 39-46: Describe the relevance of so-called “ESKAPE” bacteria, especially the Gram-negative constituents, where colistin-resistance is an important issue.
Include the following publications as a reference:
https://www.ncbi.nlm.nih.gov/pubmed/29310479
https://www.mdpi.com/2079-9721/7/2/41
https://www.ncbi.nlm.nih.gov/pubmed/19035777
Response: According to the reviewer’s comment, we modified the introduction (Lines 34-41).
Line 55: an unnecessary space after enteroaggregative
Response: We changed it (Line 122)
Results and discussion: there are three mcr-1 harboring strains tested in this study. Although this number is not that high, significant results were produced in the study. Please describe in the discussion that despite the small number of strains, why is this information novel and significant.
Response: We changed the discussion in the revised version (Lines 150-153)